# Diverse Partners of the Partitioning ParB Protein in *Pseudomonas aeruginosa*

Adam Kawalek,[a] Krzysztof Glabski,[a] Aneta Agnieszka Bartosik,[a] Dominika Wozniak,[a] Magdalena Kusiak,[a] Jan Gawor,[a] Karolina Zuchniewicz,[a] Grazyna Jagura-Burdzy[a]

[a]Institute of Biochemistry and Biophysics, Polish Academy of Sciences, Warsaw, Poland

**ABSTRACT** In the majority of bacterial species, the tripartite ParAB-*parS* system, composed of an ATPase (ParA), a DNA-binding protein (ParB), and its target *parS* sequence(s), assists in the chromosome partitioning. ParB forms large nucleoprotein complexes at *parS*(s), located in the vicinity of origin of chromosomal replication (*oriC*), which after replication are subsequently positioned by ParA in cell poles. Remarkably, ParA and ParB participate not only in the chromosome segregation but through interactions with various cellular partners they are also involved in other cell cycle-related processes, in a species-specific manner. In this work, we characterized *Pseudomonas aeruginosa* ParB interactions with the cognate ParA, showing that the N-terminal motif of ParB is required for these interactions, and demonstrated that ParAB-*parS*-mediated rapid segregation of newly replicated *ori* domains prevented structural maintenance of chromosome (SMC)-mediated cohesion of sister chromosomes. Furthermore, using proteome-wide techniques, we have identified other ParB partners in *P. aeruginosa*, which encompass a number of proteins, including the nucleoid-associated proteins NdpA (PA3849) and NdpA2, MinE (PA3245) of Min system, and transcriptional regulators and various enzymes, e.g., CTP synthetase (PA3637). Among them are also NTPases PA4465, PA5028, PA3481, and FleN (PA1454), three of them displaying polar localization in bacterial cells. Overall, this work presents the spectrum of *P. aeruginosa* ParB partners and implicates the role of this protein in the cross-talk between chromosome segregation and other cellular processes.

**IMPORTANCE** In *Pseudomonas aeruginosa*, a Gram-negative pathogen causing life-threatening infections in immunocompromised patients, the ParAB-*parS* system is involved in the precise separation of newly replicated bacterial chromosomes. In this work, we identified and characterized proteins interacting with partitioning protein ParB. We mapped the domain of interactions with its cognate ParA partner and showed that ParB–ParA interactions are crucial for the chromosome segregation and for proper SMC action on DNA. We also demonstrated ParB interactions with other DNA binding proteins, metabolic enzymes, and NTPases displaying polar localization in the cells. Overall, this study uncovers novel players cooperating with the chromosome partition system in *P. aeruginosa*, supporting its important regulatory role in the bacterial cell cycle.

**KEYWORDS** DNA segregation, *Pseudomonas aeruginosa*, partitioning proteins

Address correspondence to Adam Kawalek, a.kawalek@ibb.waw.pl, or Grazyna Jagura-Burdzy, gjburdzy@ibb.waw.pl.

The authors declare no conflict of interest.

Segregation of prokaryotic chromosomes is a multistep process involving initial separation of replicated regions proximal to the origin of chromosomal replication (*oriC*), bulk segregation of the chromosome, and separation of the termini regions (*ter*) (1). In the majority of species, the first step involves the action of homologs of the plasmidic class Ia partitioning system ParA-ParB-*parS* (2, 3), composed of ParA (an ATPase), ParB (a DNA-binding protein), and one or multiple palindromic sites recognized by ParB, designated centromere-like sequences (*parS*), mostly located in 25% of genomes around *oriC* (also called the *ori* domain) (4). ParB binding to *parS*s leads to its spreading

on adjacent DNA and formation of large nucleoprotein complexes, through a combination of 1D and 3D interactions between ParB molecules (5 to 9). The newly replicated ParB-bound *ori* domains are subsequently positioned in the opposite cell halves by ParAs, a deviant Walker type ATPases (10 to 12). Various mechanisms have been proposed to explain the directional movement of ParB-DNA complexes by ParA (12, 13). These mechanisms are based on the ability of ParA proteins to homodimerize and associate nonspecifically with DNA upon ATP binding (14 to 16). ParB–ParA interactions stimulate ATP hydrolysis, ParA monomerization, and dissociation from the nucleoid. ParA monomers may again bind ATP; however, they cannot instantly bind to DNA as a conformational change is required first. The gradient of ParA dimers attracts ParB-DNA complexes and perpetuates their relocation. Thus, the cooperation between ParB and cognate ParA is vital for the first stage of chromosome segregation.

Intriguingly, chromosomal ParA and ParB proteins were also shown to interact with other proteins in a species-specific manner (reviewed in references 3, 17, and 18). ParB-*parS* complexes serve as loading platforms for structural maintenance of chromosome (SMC) proteins aligning opposite chromosome arms and promoting DNA condensation by loop extrusion (19 to 22). The process of SMC recruitment involves direct interactions with ParB (23). Furthermore, the positioning of ParB-*parS* complexes and consequently *ori* domains in defined cell compartments may engage interactions of ParA and/or ParB proteins with specific pole-organizing proteins. Concomitantly, multiple other proteins interact with ParAB proteins, including those involved in the cell division, cell morphogenesis, cell cycle coordination, and regulation of replication initiation (3, 17, 24). A particularly interesting class of partners is represented by ParA-like ATPases with diverse functions. In *Caulobacter crescentus*, a ParB partner, the pole-organizing protein PopZ is responsible for anchoring ParB-*parS* complexes to the cell poles (25). Additionally, in this bacterium, the ParA homolog termed MipZ was shown to interact with ParB and coordinate chromosome segregation with cell division by interfering with FtsZ polymerization (26 to 28). MipZ homologs have also been characterized in *Rhodobacter sphaeroides* and *Magnetospirillum gryphiswaldense* (29, 30). In *Corynebacterium glutamicum*, ParB interacts not only with ParA but also with PldP, an ATPase with a role in division site selection (31). Remarkably, in *Streptococcus pneumoniae*, a species lacking a gene encoding the canonical ParA, a CpsD protein with ATPase activity binds to ParB to promote chromosome segregation, as well as cell division and capsule formation (32). Overall, these findings indicate species-specific ParA/ParB interactions with other cellular partners, including proteins with ATPase activity.

In *Pseudomonas aeruginosa* (here referred to as *Pae*), a rod-shaped bacterium with a simple life cycle, deletion of *parA* and/or *parB* is not lethal but results in up to 10% of anucleate cells during growth under optimal conditions (33 to 36). Concomitantly, *par* mutants exhibit longer division time, increase in cell size, and altered colony morphology and are impaired in swarming and swimming motilities, suggesting a role of Par proteins in control of various processes (34, 35). Our transcriptomic analysis showed hundreds of genes with altered expression in *Pae par* mutants (37) as well as under conditions of ParB abundance (38), suggesting direct or indirect influence of Par proteins on the transcriptome. Moreover, our recent study showed that *Pae* ParB binds not only to palindromic *parS* sequences but also to half-*parS* motifs (39).

In this work, we showed that ParA interaction with the conserved motif at the N terminus of ParB is essential for proper *ori* domain segregation. Moreover, using various proteome-wide approaches, we identified numerous novel ParB partners, including four NTPases, PA3481, PA5028, PA4465, and FleN (PA1454), with three of them displaying polar localization. Overall, our data highlight the spectrum of ParB partners in *Pae*.

## RESULTS

**Mapping the ParB region involved in the interactions with cognate ATPase partner ParA.** Chromosomally encoded ParB proteins are composed of three domains: the N-terminal domain (NTD) involved in CTP binding and hydrolysis, the central DNA-binding domain, and the C-terminal part mediating dimer formation (7, 40 to 44).

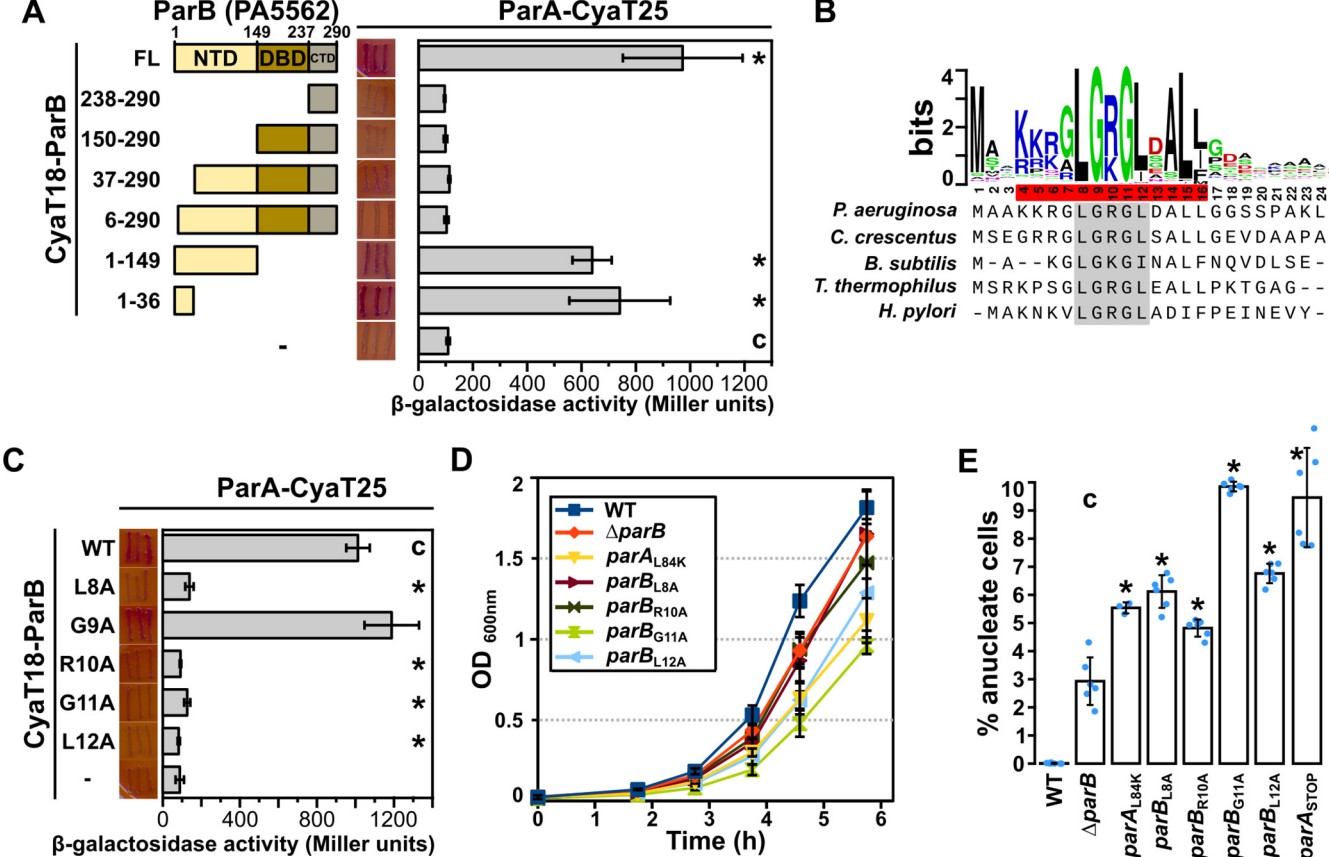

**FIG 1** N-terminal motif in ParB is required for interaction with ParA and proper chromosome segregation. (A) ParA interactions with truncated ParB variants analyzed using the BACTH system. *E. coli* BTH101 *cya*- expressing the indicated CyaT18-ParB variants and ParA-CyaT25 were streaked on MacConkey agar, photographed, and used to inoculate cultures used for $\beta$-galactosidase activity measurements. Data represent mean values from at least five cultures ± standard deviations (SD). *, $P < 0.05$ (two-sided Student's *t* test) relative to the control expressing CyaT18 and CyaT25 (indicated with "c"). Scheme illustrates the ParB truncations used. Numbers indicate ranges of amino acids, and colors indicate the domains. NTD, N-terminal domain; DBD, DNA binding domain; CTD, C-terminal domain; FL, full length. (B) Sequence logo, representing the conservation within the N-terminal part of *P. aeruginosa* ParB, obtained through iterative alignment of 822 ParB homologs, identified in representative and reference bacterial genomes, to sequence of *P. aeruginosa* ParB. Accession numbers and full-length protein alignment are shown in Fig. S1. (C) BACTH analysis of ParA interactions with the indicated ParB substitution variants. (D) Growth of *P. aeruginosa* WT and mutant strains in L broth (LB) at 37°C. Data represent mean optical density at 600 nm ± SD from 3 cultures. (E) Percentage of anucleate cells in exponentially growing cultures ($OD_{600}$ ~0.5) of PAO1161 (WT), and mutants, assessed by microscopic analysis after propidium iodide staining. Data represent mean ± SD, and blue dots represent the individual measurements of at least 2,000 cells. *, $P < 0.05$ as determined by two-tailed Student's *t* test relative to the $\Delta parB$ strain.

Previous experiments with the use of the yeast two-hybrid system suggested involvement of the *Pae* ParB C-terminal domain for the interactions with ParA (45), which was in contrast to what was acknowledged for other bacterial species. To clarify this discrepancy, here we have systematically analyzed ParA interactions with various truncated variants of ParB using the bacterial two-hybrid (BACTH) system, based on reconstitution of adenylate cyclase activity by interacting proteins fused with CyaT18 and CyaT25 subunits (Fig. 1A). The analysis showed that ParA interacts with the NTD of ParB but not with the remaining two domains (Fig. 1A). Removal of 37 or even 6 amino acids from the N terminus of ParB abrogated the interactions, and, concomitantly, the ParB$_{1-36}$ fragment interacted with ParA (Fig. 1A). This suggests that the N-terminal part of ParB is sufficient for interactions with ParA. The interactions of the dimerization domain with ParA reported previously with the use of the yeast two-hybrid system might represent a false-positive observation, possibly due to auto-activation of reporter genes, triggered by the C terminus of ParB (46).

The sequence alignment of bacterial ParBs provided a logo illuminating amino acid conservation (Fig. 1B; full logo in Fig. S1 in the supplemental material). The N terminus of ParB contains a conserved stretch of 12 amino acids with several positively charged and aliphatic amino acids. Alanine substitution of 5 highly conserved amino acids (gray

in Fig. 1B) showed that altering L8, R10, G11, and L12 but not G9 blocked the ParB interactions with ParA (Fig. 1C). The sensitivity and indirect relation between reporter activities and the extent of protein–protein interactions in BACTH could mask the residual interactions between ParA and ParB variants; nevertheless, the results clearly indicate the importance of residues at the N terminus of *Pae* ParB for interactions with ParA.

**Significance of ParB–ParA interactions in chromosome segregation.** To check the effect of these amino acid substitutions on chromosome segregation, we have engineered *Pae* PAO1161 derivatives with missense *parB* mutations resulting in production of ParB variants L8A, R10A, G11A, or L12A. Analysis of the culture growth in the rich medium showed lower growth rates of strains producing these ParB variants, similar to that observed for strains lacking *parB* or expressing ParA L84K protein, defective in interactions with ParB (36) (Fig. 1D). Interestingly, the point mutants differed in the extent of the growth retardation, and the most severe growth defect was observed for the strain producing ParB G11A (Fig. 1D). The effect seemed not to be related to the level of ParB, as it was only slightly reduced in mutants expressing ParB L8A, R10A, G11A, or L12A, compared to native ParB in PAO1161 in contrast to cells lacking ParA (Fig. S2) (33). Concomitantly, analysis of the frequency of anucleate cells in the cultures exponentially growing in rich medium showed that, whereas less than 0.1% of cells lacking nucleoid were observed for the wild-type (WT) strain and around 3% of such cells were observed for the Δ*parB* strain, the cultures of strains producing ParA and ParB variants with disrupted interactions with their cognate partners contained significantly more (5 to 10%) anucleate cells (Fig. 1E), similar to what was observed for the ParA deficient strain (33).

To check whether the ParB–ParA interaction is required for ParB complex formation on *parS*s, in the case of *Pae*, the *parS1-4* cluster adjacent to *oriC*, we have used fluorescence microscopy. In the PAO1161 strain grown in the rich medium, the majority of the cells contained up to 4 regularly spaced YFP-ParB foci (Fig. 2A, B). In the cells with two foci, they occupied positions at 20 and 80% of relative cell length (Fig. 2C, D), as reported previously (34, 46). ParB foci were still observed in strains producing interactions interactions-defective ParA L84K or ParB L8A, although they demonstrated variable intensity and irregular positioning (Fig. 2A, C, D). The number of foci detected per cell was lower, possibly due to the proximity of individual ParB-*parS1-4* complexes triggered by a defect in their separation (Fig. 2B). Similar disturbances were observed when complex localization was analyzed in cells producing ParB R10A, G11A, or L12A (Fig. S3). This confirms that interactions between ParB and ParA are not essential for ParB complex formation on *parS*s but are required for proper movements of ParB-*ori* domain complexes. Strikingly, there was a much stronger phenotypic effect of disrupted ParA–ParB interaction in comparison to the lack of ParB alone. Indeed, the phenotype of the Δ*parAB* strain was similar to Δ*parB*, suggesting that ParB not interacting with its cognate ParA is detrimental to the cells (Fig. 2E). When the crucial *parS* sequences were mutated (Δ*parS1-4* background), the presence of ParB L8A, R10A, G11A, or L12A did not further increase the anucleate cell content (Fig. 2E), confirming that impaired separation of ParB-*parS1-4* complexes was responsible for the negative effect of ParB variants (compare to Fig. 1D).

In other organisms, ParB-*parS* complexes were shown to recruit SMC, and remarkably, an asymmetry of SMC recruitment was observed recently between the two *Pae* daughter chromosomes, suggesting presence of a mechanism controlling the process of its loading on newly replicated origins (47). The detrimental effect of ParA absence or disruption of ParB-ParA interactions was largely absent in the Δ*smc* mutant, implicating that SMC could enhance adhesion of unseparated ParB-*parS* complexes (Fig. 2E). Identical observation was made when cells were grown in the defined minimal medium, slowing down the growth rate (Fig. S4). The absence of SMC did not affect ParB level (Fig. S2) and significantly restored the separation of *ori* domains in cells producing ParB G11A variant as visualized by tagging the *ori* region with an

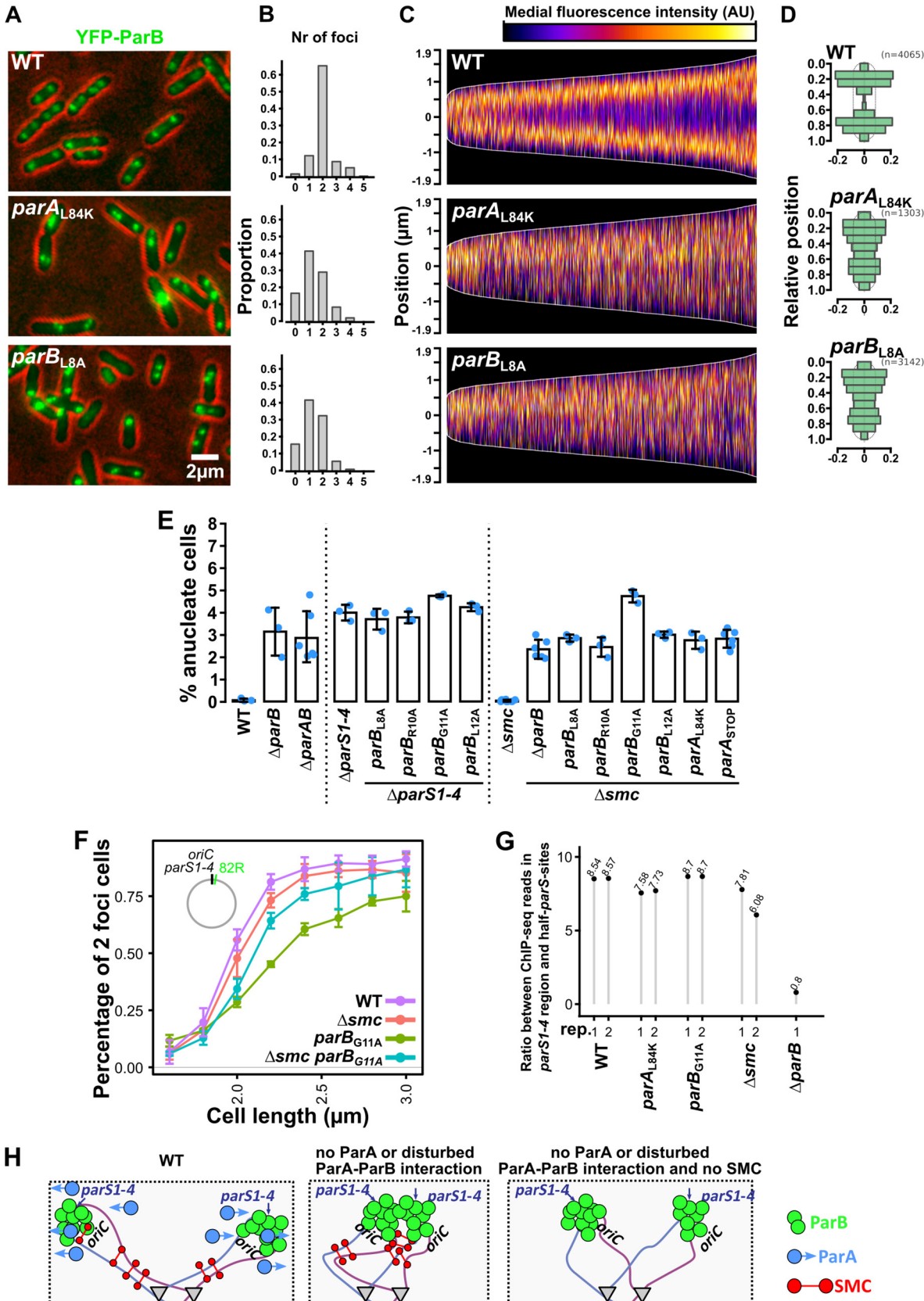

**FIG 2** Interaction of ParB with ParA is required for rapid separation of ParB-*ori* complexes, preventing chromosome cohesion by SMC. Strains carrying *tacp-yfp-parB* fusions on plasmids were grown without IPTG induction in L broth at 37°C until logarithmic phase of

orthogonal P1 ParB-*parS* system (Fig. 2F). This suggests that in the strains lacking SMC and ParA, ParB complexes could be segregated by other mechanism(s). Finally, to analyze the impact of ParB-ParA interactions and SMC on ParB binding to the chromosome, including the extent of ParB spreading around *parS*s, we performed chromatin immunoprecipitation and sequencing (ChIP-seq) in WT, $parB_{G11A}$, $parA_{L84K}$, and $\Delta smc$ strains. We calculated the ratio between the number of ChIP-seq reads in the region $\pm 15$ kb around the *parS1-4* cluster versus the number of reads mapping to sites encompassing half-*parS*s, specifically bound by *Pae* ParB (39). The use of ParB binding to the half-*parS*s, at which ParB is thought not to spread from, constitutes an internal control for each ChIP sample. Large nucleoprotein complexes formed by ParB around *oriC* were observed in all analyzed mutants, in agreement with the fluorescence microscopy analyses. Concomitantly, there was no significant effect of disrupted ParB–ParA interactions (ParB G11A or ParA L84K) or SMC presence on the ratio, suggesting no major impact of these factors on the ability of ParB to remain bound to the DNA within the complex around *parS1-4* (Fig. 2G). Overall, this suggests that ParB interactions with ParA mediated by a conserved motif at the ParB N terminus are required only for rapid partition of newly replicated *ori* regions, which prevents their cohesion by SMC (Fig. 2H).

**Identification of novel ParB-interacting proteins in *P. aeruginosa*.** The complex phenotype of cells lacking the partitioning proteins in *P. aeruginosa* suggested their regulatory character (33, 34, 37, 38) and prompted the analysis of the ParB protein interaction network beyond ParA. The candidate ParB-interacting proteins were identified using three screening methods: (i) coimmunoprecipitation (CoIP) with the use of antibodies targeting ParB followed by identification of associated proteins by mass spectrometry; (ii) using ParB as a bait in the bacterial two hybrid screening of the *Pae* genomic library; and (iii) analysis of ParB interactions with proteins showing sequence similarity to ParA (Fig. 3A).

The CoIP analysis performed with the use of anti-FLAG antibodies and extracts from PAO1161 cells producing FLAG-ParB, showed 250 candidates among which 40 proteins were found exclusively in one or more CoIP samples and not in controls (Table S1). These encompassed SMC (PA1527) and its partner ScpB (PA3197). The remaining 210 candidate proteins were also detected in the corresponding control samples, although with significantly lower scores. These were kept in the analysis as this group encompassed ParA and ParB itself (Table S1). Among them, multiple transcriptional regulators were found, proteins involved in the genome integrity and topology maintenance (e.g., UvrD, GyrA, GyrB, ParC, and ParE), nucleoid associated proteins (NAPs) (e.g., HupB, PA1533, and NdpA [PA3849]), as well as proteases ClpP, ClpP2, Lon, and various enzymes (Table S1).

Verification of the interactions between ParB and selected candidate partners, iden-

**FIG 2** Legend (Continued)

growth ($OD_{600}$ of 0.5), and cells were analyzed using fluorescence microscopy. (A) Representative microscope images showing cellular localization of YFP-ParB in wild type and mutants producing ParA L84K or ParB L8A. Contrast was enhanced and cell contour was false-colored in red. (B) Distribution of the number of YFP-ParB foci per cell in the analyzed strains. Foci were detected in fluorescence images using MicrobeJ with the same criteria for all strains. The differences between mutant strains and WT are significant (Chi-squared goodness of fit test, $P < 1 \times 10^{-5}$). (C) Distribution of YFP-ParB fluorescence along the long axis of the cells. Profiles of YFP-ParB fluorescence of individual cells were sorted based on cell length, and oriented in a way in which the top (first pole) represents the cell half with higher mean fluorescence signal. (D) Relative position of YFP-ParB foci analyzed in two foci-containing cells. Each cell was split in 10 bins along the long axis, and the histogram was mirrored. (E) Percentage of anucleate cells in exponentially growing cultures ($OD_{600}$ ~0.5) of the indicated strains. Data represent mean $\pm$ SD, and blue dots represent the individual measurements. At least 2,000 cells were analyzed for each repeat. (F) Analysis of *ori* region separation in the indicated strains. Cells harboring $parS_{P1}$ located at position 82R (82 kb on the right chromosome arm) and expressing $ParB_{P1}$-GFP were grown in M9 medium with glucose at 37°C (conditions of slow division) and analyzed by fluorescence microscopy. Data represent percentage of two $ParB_{P1}$-GFP foci cells according to cell size. At least 3,000 cells were analyzed for each repeat, and data represent mean $\pm$ SD for three cultures. (G) Impact of interaction of ParA and SMC presence on the extent of ParB spreading. ChIP-seq analysis was performed using anti-ParB antibodies and indicated strains. Sequencing reads were mapped to the PAO1161 reference genome. Data represent the ratio between the number of reads mapping to the *parS1-4* $\pm$ 15 kbp region and to other ParB ChIP-seq peaks (encompassing half-*parS* sequences as previously demonstrated [39]) for individual ChIP-seq replicates. (H) Model explaining the deleterious impact of SMC in strains with no ParA or disturbed ParB-ParA interactions. In WT cells, rapid separation of newly replicated regions with *oriC* prevents SMC-mediated clamping of sister chromosomes. Apparently, other mechanisms must be involved in separating ParB-*parS* complexes (and *ori* regions), in the absence of ParA (or when the interactions between ParB and ParA are disturbed) and when SMC is not present.

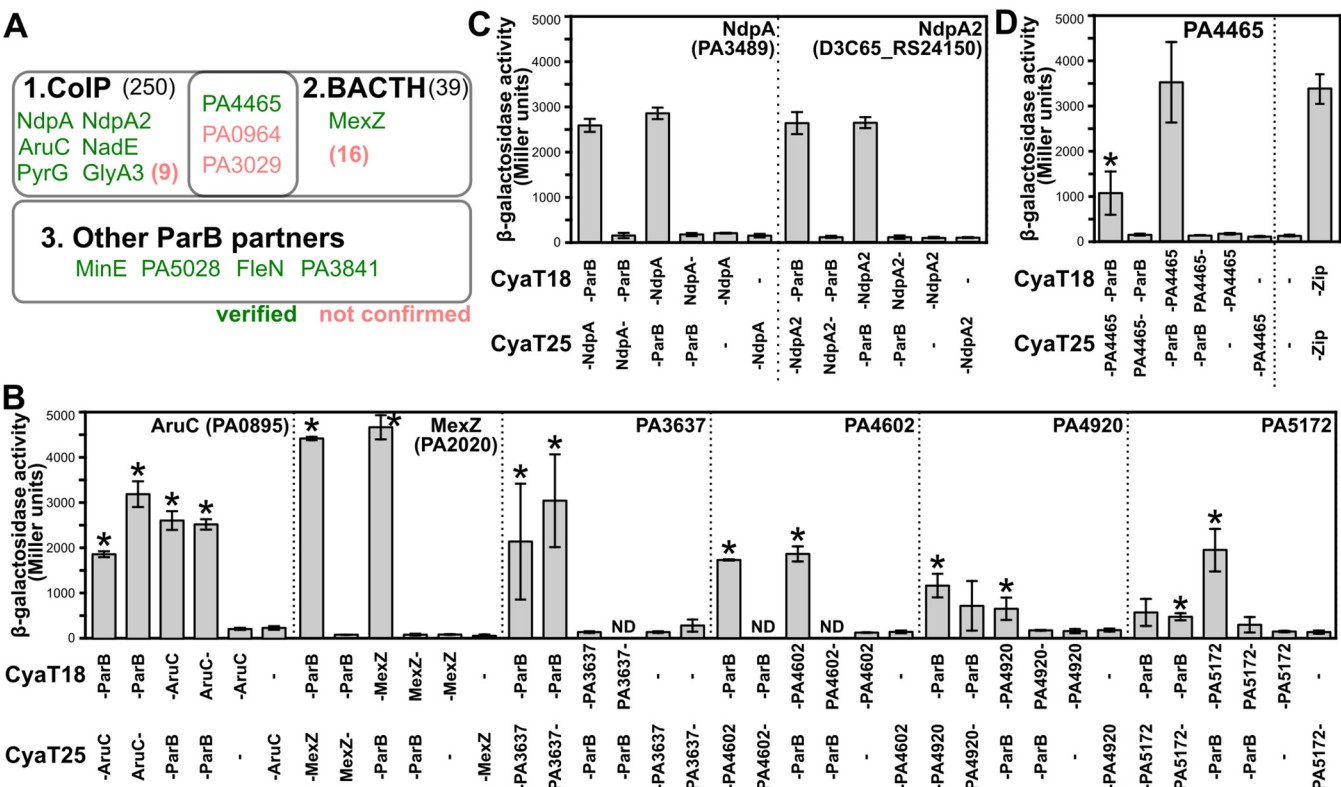

**FIG 3** Identification of novel ParB-interacting proteins in *P. aeruginosa*. (A) An overview of the outcome of three approaches used in this study for identification of ParB partners. Colors indicate the proteins for which the interactions were or were not confirmed in the BACTH system using full-length proteins. (B to D) BACTH analysis of the interactions between ParB and indicated proteins. *E. coli* BTH101 *cya-* transformants expressing the indicated proteins fused to CyaT18 or CyaT25 (dash indicates the junction) were used to inoculate cultures to perform β-galactosidase activity assays. Data represent mean activity from at least five cultures ± standard deviations (SD). *, $P < 0.05$ (as determined by one-tailed Student's *t* test relative to corresponding control cultures—strains with only one fusion protein).

tified in the CoIP by BACTH, confirmed several enzymes as ParB partners (Fig. 3B). These included AruC (PA0895), a part of the catabolic arginine succinyltransferase pathway (48), as well as *Pae* CTP-synthase PA3637 (PyrG), and others (Table 1, Fig. 3B). While it is tempting to speculate that ParB complexes are involved in control of subcellular localization of metabolic activities, further studies with individual proteins are required to address this relation. An intriguing candidate partner of ParB, identified by CoIP and confirmed by BACTH (Fig. 3C), was NdpA/Yejk (PA3849) protein, representing

**TABLE 1** Novel ParB partners analyzed in this work. The screening was based on coimmunoprecipitation (Table S1), screening of ParB-interacting proteins using *P. aeruginosa* genomic library in bacterial two-hybrid vector (Table S2), as well as the targeted approach (Fig. 4A)

| PAO1 ID | Protein | Product | Identified using: |
|---|---|---|---|
| PA0895 | AruC | N2-succinylornithine 5-aminotransferase (SOAT) | CoIP |
| PA3637 | PyrG | CTP synthetase | CoIP |
| PA3849[a] D3C65_24125 | NdpA/Yejk NdpA2 | Nucleoid-associated protein | CoIP |
| PA4602 | GlyA3 | Serine hydroxymethyltransferase | CoIP |
| PA4920 | NadE | NH3-dependent NAD synthetase | CoIP |
| PA5172 | ArcB | Ornithine carbamoyltransferase | CoIP |
| PA2020 | MexZ | TetR family transcriptional regulator | BACTH |
| PA4465 | YhbJ/RapZ | P-loop containing protein | CoIP, BACTH |
| PA3481 | Mrp | Conserved hypothetical protein | Targeted |
| PA5028 | | Conserved hypothetical protein | Targeted |
| PA1464 | FleN | Flagellar synthesis regulator | Targeted |
| PA3245 | MinE | Cell division topological specificity factor | Targeted |

[a]PAO1161 genome additionally encodes NdpA2 protein, which also interacts with ParB (Fig. 3C).

a poorly characterized class of nucleoid associated proteins (49, 50). Interactions with ParB were also observed for the paralogous NpdA2 protein (Fig. S5A, Fig. 3C). NdpA2 is encoded within PAPI-1 family integrative conjugative elements present in genomes of strains like PA14 or PAO1161 but not in PAO1. Analysis of strains lacking *ndpA*, *ndpA2*, both genes, or the strain overproducing NdpA did not show defects in growth or elevated amount of anucleate cells, suggesting that under conditions tested, NdpA proteins may have a function unrelated to chromosome segregation (Fig. S5B, C).

A genome-wide BACTH-based screening was also performed with the use of the vector encoding CyaT25-ParB as a bait and the *Pae* genomic library cloned into the compatible plasmid, allowing fusion of random fragments to CyaT18 (51), which resulted in 37 candidate proteins (Table S2). These encompassed various N-terminal truncations of ParB protein, validating the approach, as the genomic library construction protocol allowed mostly C-terminal regions of prey proteins to be fused with CyaT18, and the C-terminal domain of ParB mediates protein dimerization (52). A large fraction (24%) of these were the known or putative transcriptional regulators. For most of the partners identified by BACTH screening, the interactions could not be confirmed when full-length candidate proteins were used in the BACTH assays (Table S2), with the exception of MexZ (PA2020), TetR family transcriptional regulator being the repressor of multidrug efflux pump gene *mexXY* in *Pae* (53) (Fig. 3B).

Three proteins were identified in both CoIP and BACTH screenings: molybdopterin biosynthetic protein B2 (MoaB2, PA3029), PqsR-mediated PQS regulator PmpR (PA0964), as well as PA4465, but only for the PA4465 or the last one the interactions between ParB and full-length protein were observed with BACTH (Fig. 3D). The sequence of the *Pae* PAO1 PA4465 protein shows 50% identity (99% coverage) to *E. coli* K-12 RNase adapter protein RapZ (UniProt, P0A894) and 36% identity (99% coverage) with *B. subtilis* YvcJ (UniProt, O06973), two proteins that were previously shown to display ATPase and GTPase activities (54 to 56). This suggested that a protein with an NTPase activity, other than ParA, may interact with ParB in *Pae*.

Finally, in the targeted approach, we used BACTH to probe ParB interactions with proteins showing sequence homology to ParA (Fig. 4A). These encompassed PA3244, encoding a homolog of *E. coli* MinD protein, a component of the *Pae* MinCDE system (57). The construction and analysis of *Pae min* mutants confirmed the predicted function of these genes since their deletions resulted in the extreme cell elongation and presence of minicells (Fig. 4B, C). The systematic analysis of interactions between the Min and Par proteins system revealed self-interactions of MinC, MinD, and MinE and the interactions between MinD-MinC and MinD-MinE, but no interactions between MinC and MinE, similarly as for *E. coli* homologs (58). Despite the ParA and MinD similarities, no interactions with any of the Par proteins were detected (Fig. 4D). Interestingly, MinE interacted with ParB, whereas MinC interacted with ParA (Fig. 4D), suggesting an intricate cross-talk between Par and Min systems in *Pae* (Fig. 4E).

Probing ParB interactions with the four other proteins showing sequence homology to ParA (Fig. 4A) revealed three other partners: PA3481, PA5028, and FleN (PA1454) (Fig. 4F). PA3481 is a Mrp/ApbC/NBP35 subfamily putative NTPase, with 48% amino acid sequence identity to *E. coli* Mrp, possibly involved in metabolism of Fe–S clusters. PA5028 is an uncharacterized orphan P-loop NTPase, and FleN (PA1454) is a regulator of the expression of flagellar genes in *Pae* (59, 60). Importantly, interactions between ParB and these three proteins were also confirmed using CoIP, when corresponding protein pairs were produced in a heterologous host *E. coli* (Fig. 4G). The fourth ParA homolog, PA1462, was identified as a candidate ParB partner in CoIP analysis (Table S1); however, in the BACTH analysis, the interactions between fusion proteins were not confirmed (Fig. 4F). Overall, these data indicate that at least four proteins with presumed NTPase activity, PA4465, PA3481, PA5028, and FleN, may interact with partitioning protein ParB from *Pae*.

**Significance of newly discovered ParB partners in the cell growth and chromosome segregation.** The newly identified ParB partners, PA4465, PA3481, PA5028, and FleN, do not compete with ParA for ParB binding, as the analysis of their interaction

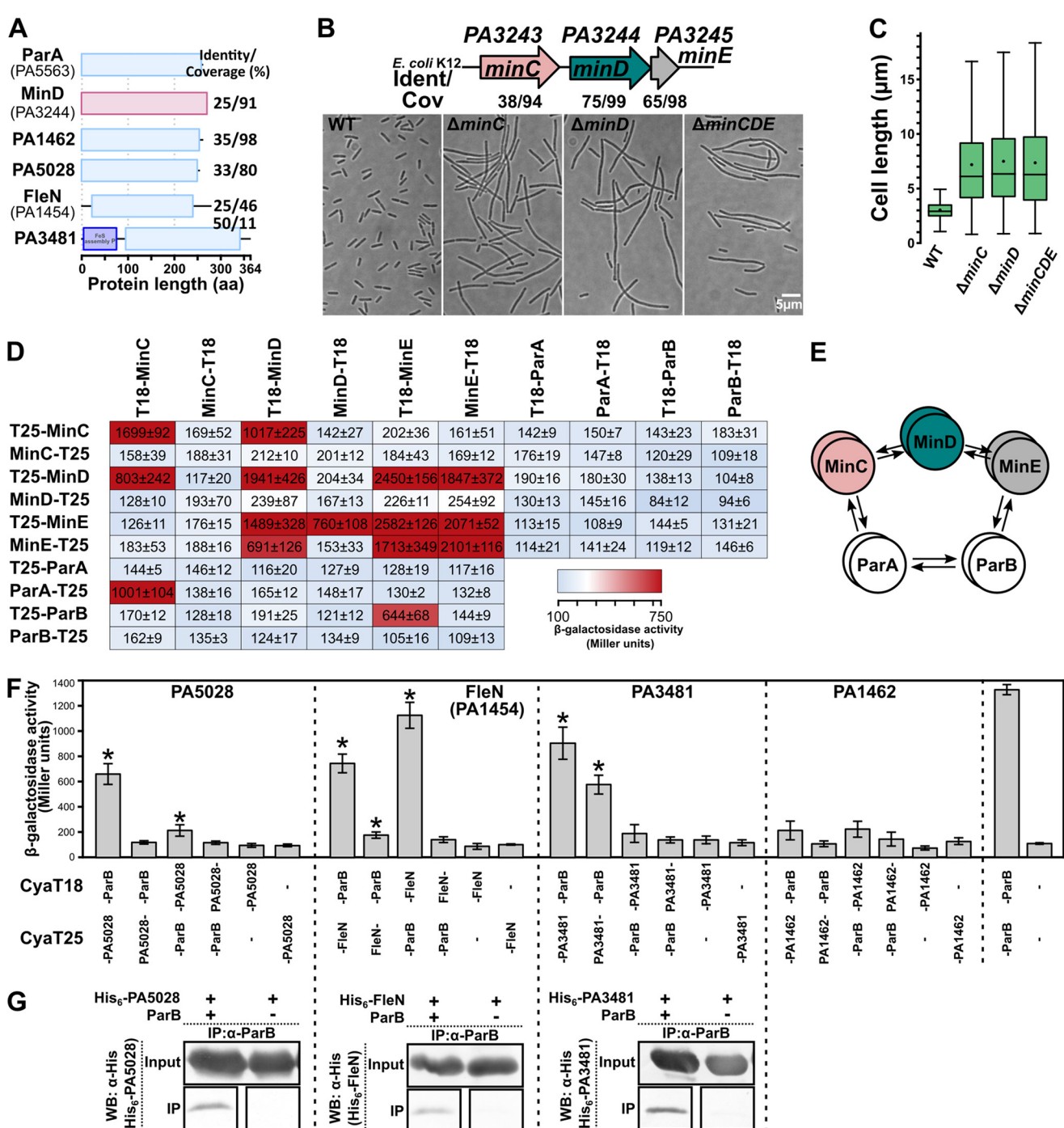

**FIG 4** ParB interacts with MinE as well as PA5028, FleN (PA1454), and PA3481—three proteins showing amino acid sequence similarity with ParA. (A) *P. aeruginosa* PAO1 proteins showing significant sequence identity with ParA. (B) Functional analysis of *P. aeruginosa* Min system. Numbers indicate amino acid identity with *Escherichia coli* K-12 proteins MinC (WP_001301105.1), MinD (WP_000101055.1), and MinE (WP_001185665.1). Impact of the lack of *min* genes on morphology of *P. aeruginosa* cells was analyzed using microscopy. Strains were grown in L broth to $OD_{600}$ of 0.3 and fixed. Representative bright-field images are shown. (C) Cell size distribution for indicated strains. Cells were measured on microscopic images using MicrobeJ (97). (D) BACTH analysis of the MinCDE and ParAB interactions. *E. coli* BTH101 *cya-* transformants expressing the indicated protein fusions were used to inoculate cultures for $\beta$-galactosidase activity measurements. Data represent mean activity from at least three cultures ± standard deviations (SD). Background activity was similar in all experiments (CyaT18/CyaT25, ~130 U), whereas control cells producing fusions CyaT18-Zip and CyaT25-Zip showed activity higher than 2,500 U. (E) Schematic presentation of the interactions between Min and Par proteins in *P. aeruginosa*. (F) BACTH analysis of the interactions between proteins with sequence similarity to ParA. $\beta$-galactosidase activity was analyzed in *E. coli* BTH101 *cya-* transformants producing the indicated proteins fused to CyaT18 or CyaT25 (dash indicates the junction). Data represent mean values from at least six cultures ± standard deviations (SD). *, $P < 0.05$ as determined by one-tailed Student's *t* test relative to corresponding control cultures (strains with only one fusion protein). (G) Analysis of the ParB-partner protein interactions using coimmunoprecipitation. $His_6$-tagged proteins were produced in *E. coli* BL21 (DE3) cells together with WT ParB, cross-linked with formaldehyde and subjected to the immunoprecipitation procedure with the use of anti-ParB antibodies. Western blot analysis using anti-$His_6$ antibodies was performed on both cell extracts (input) and samples after immunoprecipitation (IP).

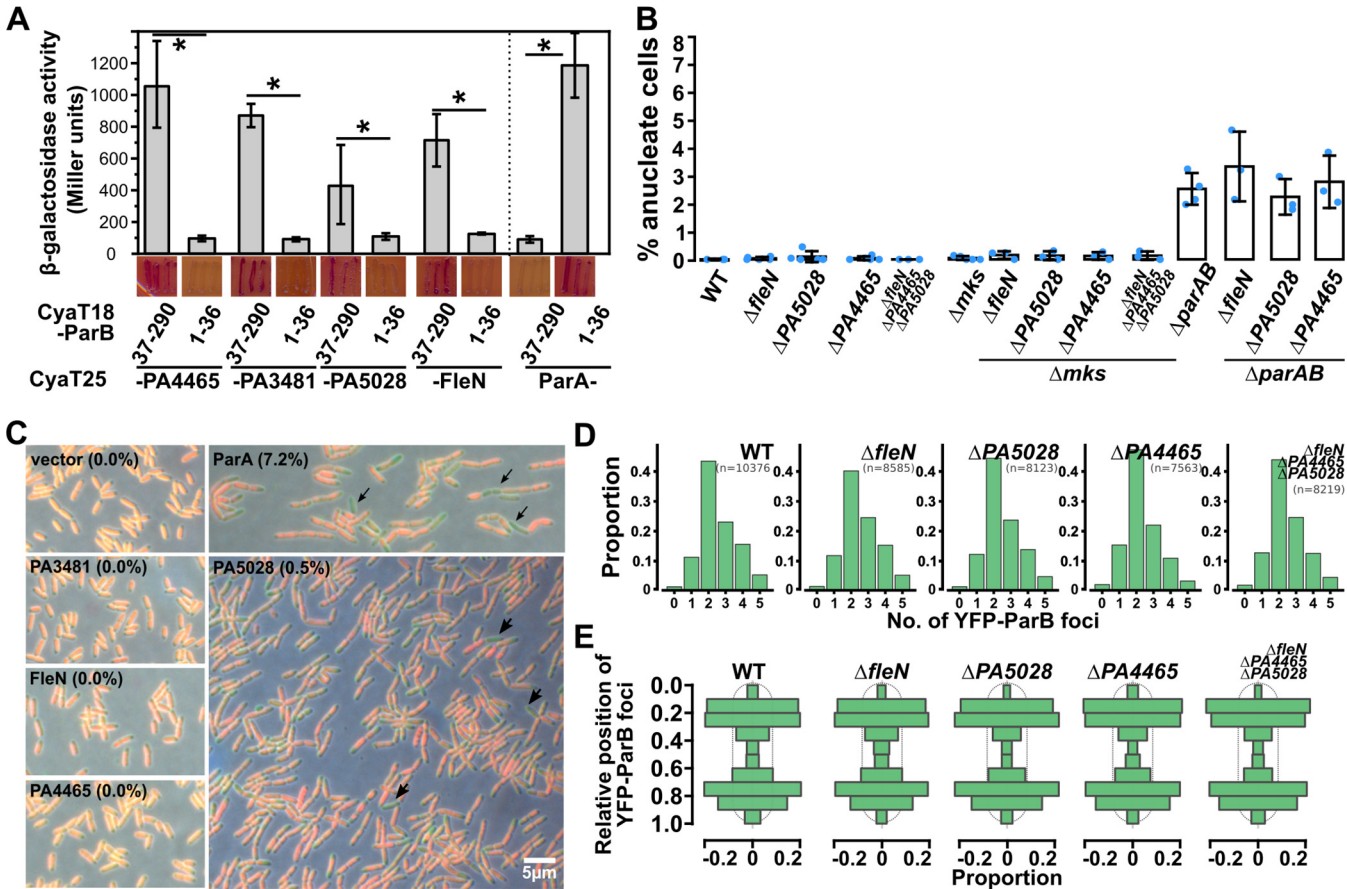

**FIG 5** The ParB-interacting putative NTPases do not play a major role in DNA segregation. (A) BACTH analysis of the interactions between two ParB fragments: ParB$_{37-290}$ and ParB$_{1-36}$ and the putative NTPases. *E. coli* BTH101 *cya*- expressing the indicated fusions were streaked on MacConkey agar, photographed, and used to inoculate cultures to perform $\beta$-galactosidase activity assays. Data represent mean values from at least six cultures $\pm$ standard deviations (SD). *, $P < 0.05$ in two-sided Student's *t* test. (B) Percentage of anucleate cells in exponentially growing cultures (OD$_{600}$ ~0.5) of the indicated strains. Data represent mean $\pm$ SD, and blue dots represent the individual measurements (>2,000 cells were analyzed for each culture). (C) Microscopic analysis of DNA content in strains overproducing the indicated proteins. Cells were grown in L broth with 0.1 mM IPTG, inducer of *tac*p, fixed, and stained with propidium iodide (red) and SYTO9 (green). Arrows indicate cells without nucleoid. Numbers in brackets indicate mean anucleate cell content calculated for at least 1,000 cells from three cultures. (D) Distribution of the number of YFP-ParB foci per cell in the analyzed strains. Cells producing YFP-ParB were grown in L broth (OD$_{600}$ of 0.3) at 37°C, fixed, and analyzed by fluorescence microscopy. Foci in each cell were detected using MicrobeJ. Histograms represent combined data for cells from three cultures. (E) Impact of indicated gene deletions on ParB complex localization in the cells. YFP-ParB foci were assigned to one of the 10 cell segments along the long axis. Histograms represent relative YFP-ParB foci position in two foci-containing cells of indicated strains.

with ParB 1 to 36 and 37 to 290 fragments showed that for all of them, the region of ParB involved in the interactions is different than for ParA (Fig. 5A). To check the significance of these proteins in *Pae* and their cooperation with ParB, we analyzed (i) the impact of lack or excess of these proteins on culture growth and chromosome segregation and (ii) the impact of a partner absence on the other protein cellular distribution. Despite using various approaches, a PAO1161 Δ*PA3481* mutant could not be constructed. Previously, the genes encoding PA3481 orthologs (e.g., PA14_19065) were indeed listed as essential in transposon mutagenesis analysis (61).

The growth analysis of Δ*PA5028*, Δ*fleN*, or Δ*PA4465* mutants showed no differences relative to the parental PAO1161 (WT) strain (Fig. S6A). The absence of these genes did not result in the appearance of anucleate cells in the mutant cultures (Fig. 5B) and did not alter anucleate cell content in populations of strains carrying additional deletion of *parAB* genes (Δ*parAB*, Fig. 5B). The condensin complex MksBEF may support chromosome segregation in the absence of a functional ParAB-*parS* system in the *Pae* (47, 62), hence it was hypothesized that minor defects of the ParAB-*parS* system (caused, e.g., by lack of partner proteins) could be masked by the action of MksBEF. However, deletion of *PA4465*, *PA5028*, *fleN*, or all three of them in PAO1161 Δ*mksBEF* also did not result in the appearance of anucleate cells in cultures (Fig. 5B).

The analysis of the growth of strains overproducing the proteins showed a negative effect of PA5028 and FleN excess on culture growth, whereas no impact was noted in the case of PA3481 and PA4465 (Fig. S6B). Interestingly, the excess of PA5028 frequently resulted in formation of cell chains, cells with guillotined nucleoids (Fig. 5C), and the increase in the number of anucleate cells (up to 0.5%). The absence of any of these proteins, however, did not affect the cellular distribution of ParB complexes as well as the numbers of detectable YFP-ParB foci (Fig. 5D, E). This suggests that analyzed proteins do not play a major role in *Pae* chromosome segregation under the conditions tested. Nevertheless, the observation that PA5028 abundance affects both cell morphology and DNA distribution suggests a role of this protein in the *Pae* cell cycle.

**PA3481, PA5028, and PA4465 display polar localization in *P. aeruginosa* cells.** To address the impact of ParB on the cellular distribution of ParB partners, fluorescence microscopy analyses were performed. The ParA-CFP fluorescence signal displayed an asymmetric distribution, often showing comet-like profiles along the long axis of the cells in the parental (WT) strain (Fig. 6A). Quantitative analysis of the fluorescence distribution asymmetry, analyzed by calculation of the ratio between mean fluorescence signal in each half of the cell, confirmed asymmetric distribution of ParA-CFP as the signal was more unevenly distributed than in cells with a free fluorescent protein (sfGFP) or not expressing such a protein (control for auto-fluorescence, Fig. 6B). Concomitantly, lack of ParB led to a less smooth ParA-CFP signal, possibly due to ParA association with the nucleoid (Fig. 6A). Reduced asymmetry of ParA-CFP signals in the cells was observed in Δ*parB* as well as in the strain with mutated *parS1-4* sequences (Fig. 6B). The strains with disrupted ParA–ParB interactions displayed various degrees of ParA-CFP asymmetry (Fig. 6B). These data confirm that ParB affects the cellular distribution of its cognate ATPase partner, ParA. A similar analysis with the other identified NTPase partners showed that FleN was equally distributed in the cells independent of the terminus to which fluorescent protein was attached (Fig. S6). The other three proteins showed discrete distribution in the cells. YFP-PA3481 could be observed in the majority of the cells as one or two foci at cell poles (Fig. 6C, D), and the number of foci increased with the cell length (Fig. 6E). Identical distribution and foci numbers were observed when the analysis was performed in ParB-deficient cells (Fig. 6F). Similar analysis of YFP-PA5028 showed that for this protein the signal is mostly diffused in the cells; however, also 1 to 4 discrete foci could be observed (Fig. 6G, H). These were mainly localized at cell poles, but foci at mid-cell were also observed (Fig. 6I). The foci number and positioning were independent of ParB (Fig. 6H, I). Finally, sfGFP-PA4465 foci were also observed in ∼7% of cells, and they were located at cell poles (Fig. 6J, K, L), and such distribution was not dependent on ParB. Overall, this indicates that at least three putative NTPases interacting with *Pae* ParB may be located at the cell poles.

## DISCUSSION

In this work. we have identified putative proteins interacting with partitioning protein ParB in *P. aeruginosa*. In this bacterium, both ParB and the cognate ATPase ParA are required for chromosome segregation (33, 34, 45, 63, 64); however, the molecular basis of the interactions between the two proteins in this species was not clear. Here, we showed that mutations L8A, R10A, G11A, or L12A in the conserved motif at the N terminus of ParB disrupted the interactions with ParA and disorganized the chromosome segregation process (Fig. 1), which is in agreement with studies in other organisms showing the docking site for ParA molecules in the N-terminal region of ParB (16, 65). The intracellular mobility of the ParAB-*ori* complex is thought to be driven by a processive interaction between ParA and ParB proteins, and in plasmidic class Ia partitioning systems, cognate ParB enhances ParA ATPase activity (16, 65 to 70). For Soj (ParA) from *Thermus thermophilus*, its ATPase activity is stimulated by Spo0J (ParB) as well as its N-terminal 20 amino-acid region, and R10A mutation abrogates the increase of ATPase activity (16). Similarly, mutations in K3 and K7 of Spo0J (ParB) from *B. subtilis* abolish the potentiation of Soj (ParA) activity (71). The ParA-interacting motif is connected to the rest of the

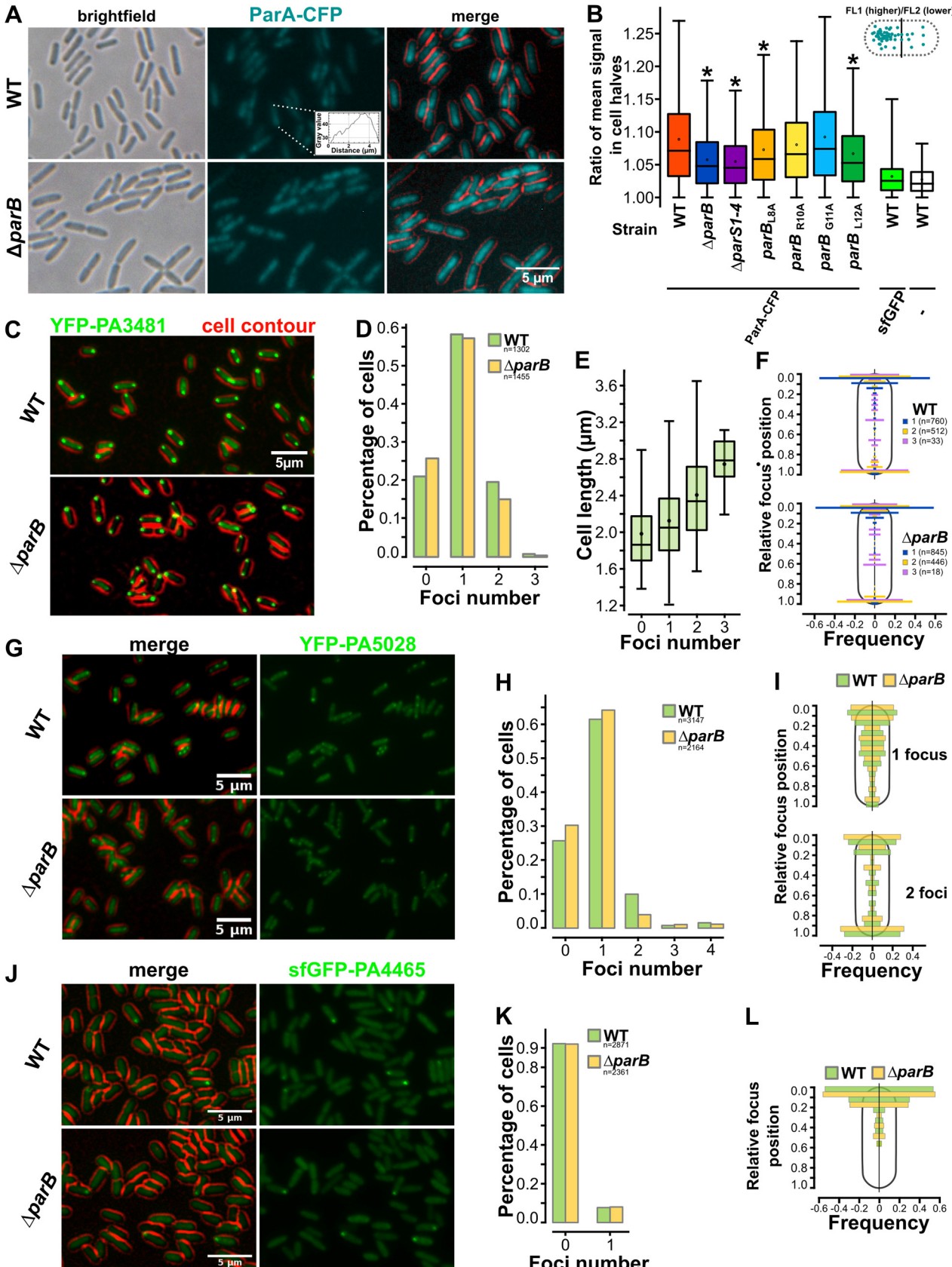

**FIG 6** Lack of ParB influences the cellular distribution of ParA but not PA4465, PA3481, or PA5028. (A) Cellular distribution of ParA-CFP in WT and Δ*parB P. aeruginosa* strains. Inset represents fluorescence signal intensity profile for the indicated cell. Cell contour was false-colored in red.

ParB protein by a flexible linker facilitating "fly-fishing" for ParA molecules (41). The data presented here show that these features are likely to be conserved in *Pae* ParB. The nucleoprotein complexes are still formed by ParB in the absence of ParA or when the ParB–ParA interactions are disrupted (Fig. 2A to D), and the ChIP-seq analysis indicated no major effect of ParA on the ParB binding to DNA and range of spreading around the *parS1-4* cluster (Fig. 2G). The main factor affecting the size of ParB partition complexes appears to be the CTPase activity of ParB (7, 41, 72, 73), and our data suggest that ParA may not affect this activity of ParB and is therefore only required for positioning of the complexes.

The involvement of ParB-*ori* interactions in SMC recruitment is well established (19, 22, 62), and in our wide search for ParB partners, SMC and its partner ScpB were found in the CoIP-based screening. Recently, an asymmetry of SMC recruitment was observed between the two daughter chromosomes of *Pae*, suggesting a mechanism limiting the process of SMC loading initially to only one newly replicated origin (47). Here, we observed that the presence of SMC in the cells with unseparated ParB complexes is detrimental for the cells, possibly due to the cohesion of the newly replicated origins (Fig. 2H). The extent of the negative effect varied depending on the ParB mutant used, and this effect could simply be a result of the extent of defect in ParA–ParB interactions. Another possibility is, however, a differential impact of mutations in the N-terminal part of ParB on an additional role of the ParA protein (e.g., in the regulation of SMC dynamics) (74). Thus, this indicates that rapid ParAB-driven *oriC* segregation, and a delay in the loading of SMC on one of the replicated origins, works in concert to prevent SMC-mediated cohesion of the newly replicated chromosomes.

The screenings used in this work uncovered numerous novel proteins interacting with ParB (Table 1). Those encompass metabolic proteins AruC (PA0895), an N2-succinylornithine 5-aminotransferase (48), GlyA3 (PA4602), NadE (PA4920), and ArcB (PA5172) (Fig. 3B). Moreover, interactions were observed with PyrG (PA3637) encoding a CTP synthetase (75). Since CTP binding and hydrolysis were recently shown to be crucial for ParB functions (7, 42, 72, 76, 77), it is tempting to speculate about a functional relevance of such interactions. Our analysis also demonstrated ParB interactions with NdpA and its paralogue NdpA2 (Fig. 3C). NdpA proteins are encoded in the majority of bacterial chromosomes and on mobile genetic elements, including integrative and conjugative elements (78 to 80). NdpA2 encoded on PAPI-1 acts in synergy with a local regulator TprA, removing a repressive mechanism exerted by the MvaT (H-NS) protein to stimulate conjugative transport of the element (78). A recent report indicated that *E. coli* YejK (NdpA homolog) interacts with both DNA gyrase and topoisomerase IV and influences their activities (50). Interestingly, both gyrase subunits, GyrB (PA0004) and GyrA (PA3168), as well as topoisomerase IV subunits ParE (PA4967) and ParC (PA4964), were among candidate ParB partners identified in Co-IP analysis (Table S1). Thus, its plausible that NdpA might be cooperating with these proteins to resolve topological problems following *oriC* replication, as suggested for topoisomerase I from *Streptomyces coelicolor* (81). Further studies should identify the partners of NdpA as well as *npdA* genetic interactions, as it is plausible that its biological role might be redundant with other nucleoid associated proteins.

During this work, the interplay between the Min system and Par proteins of *Pae* has been discovered. The interactions between these two pattern-forming protein systems are intriguing since one provides the spatiotemporal mechanisms of positioning the

**FIG 6** Legend (Continued)

(B) Analysis of the fluorescence asymmetry in various *P. aeruginosa* strains expressing the indicated fluorescent fusion proteins or sfGFP alone. Following image acquisition, the cell edges were detected using bright-field images, and fluorescence signals were quantified on the corresponding fluorescence images. The cell was split in half (transverse plane), and the fluorescence in each half was calculated. Data represent the distribution of the ratio between the higher and lower signal of each cell half. * indicates a significant difference ($P < 0.001$) in the Mann-Whitney U Test relative to WT. Fluorescence microscopy analysis of WT and Δ*parB* strains producing fluorescent protein fusions (exponentially growing in L broth with 0.01 mM IPTG at 37°C). Fluorescence microscopy images showing cellular distribution of (C) YFP-PA3481, (G) YFP-PA5028, and (J) sfGFP-PA4465 in PAO1161 (WT) and Δ*parB* cells. Distribution of the number of (D) YFP-PA3481, (H) YFP-PA5028, and (K) sfGFP-PA4465 foci per cell in WT and Δ*parB* strains. (E) Relation between YFP-PA3481 foci number and cell length. Analysis of the impact of ParB deficiency on cellular position of (F) YFP-PA3481, (I) YFP-PA5028, and (L) sfGFP-PA4465 foci. Histograms represent relative foci position in the cells with indicated number of foci. Foci in each cell were detected using MicrobeJ and assigned to one of the 10 cell segments along the long axis of the cells.

division machinery, and the other drives the chromosome positioning in the cells. Significantly, ParAB proteins interact with various components of the MinCDE machinery, ParB interacts with MinE, and MinC is a partner of ParA. *Pae* MinCD proteins organize into filaments at the membranes apart from the midcell, and they may form the scaffold for other proteins to bind there (82). Interactions with Min proteins may facilitate positioning of ParB-*ori* complexes in the cells and/or coordinate the cell division process with chromosome segregation (3).

ParA and/or ParB interactions with a hub-like protein in Gram-positive bacteria called DivIVA/Wag31, equivalent to MinE, have been reported in several strains (3). What is more, in *S. coelicolor*, ParA interacts with the coiled-coil protein Scy, an element of the tip-organizing complex also encompassing DivIVA to anchor the segrosome at the tips (83). Similarly, in *Myxococcus xanthus*, ParA binds to ParB-like protein PadC, which is required for recruitment of inactive ParA molecules to the cell pole-associated bactofilin cytoskeleton (84). In *Vibrio cholerae*, ParAI, involved in segregation of chromosome I, binds with the polar localized protein HubP (85). In *C. crescentus*, the pole-organizing protein Z (PopZ) is thought to assemble a porous homo-polymeric matrix that captures the ParB-*parS* complex at cell poles via interactions with ParA and ParB (25, 86 to 90). Thus, a great range of mechanisms is utilized by different bacteria to anchor ParB-*parS* complexes, and the proteins identified in this study might be a part of such a system in *Pae*.

The screening of the BACTH library and Co-IP pinpointed various ParB partners with NTPase activities. The PA3481, PA4465, and PA5028 displayed enhanced polar localization in fluorescence microscopy analysis (Fig. 6). Interestingly, PA3481, with similarity to the Mrp/ApbC/NBP35 subfamily of proteins involved in metabolism of Fe–S clusters (91), displayed a clear polar localization in almost all cells (Fig. 6C). Previous analyses with *Desulfovibrio vulgaris* Mrp$_{ORP}$ also showed that this protein localized to one or two poles (54, 92). The PA4465 homolog in *B. subtilis*, YvcJ, was shown to be involved in competence regulation (55, 56). YvcJ can be localized in the cell in a helical pattern or as foci close to the poles depending on the stage of growth (56). The third NTPase with a polar localization, PA5028 has not been thoroughly characterized; however, PA5028 mutants were found in PCR-based signature-tagged mutagenesis of mutants showing limited invasiveness (93). The identification of polar proteins as ParB partners suggests that akin to other bacteria, *Pae* ParB complexes might be anchored at specific polar positions in cell halves. Since the analysis excluded the individual roles of PA4465 and PA5028 in ParB-*ori* localization, they may be achieved by multiple overlapping mechanisms in *Pae*. Other approaches will be used to lower the expression of essential PA3481 protein to elucidate its role in this process. Another plausible explanation for the relevance of detected interactions is that ParB influences the NTPases activity of the partners, akin to ParA, and this activation is beneficial if happening at a particular compartment or moment of the cell cycle. Our data indicate that the abovementioned ATPases might not directly compete with ParA for ParB binding, as their binding does not involve the motif at the N terminus; nevertheless, we might not rule out the impact of protein partners on, e.g., ParB-ParA interactions, DNA binding by ParB, or CTP processing.

Overall, this study pointed out the possible involvement of partitioning the ParB protein of *Pae* in the cell cycle by interacting with the proteins controlling the cell division and possibly spatial organization of the cells. It also demonstrated ParB's role in the metabolic activity of the cells through direct interactions with the various enzymes, but also the proteins that sculpture bacterial chromosome and influence its topology. Further studies should shed light on the molecular basis and significance of interactions with individual partners.

## MATERIALS AND METHODS

**Bacterial strains and growth conditions.** Strains, plasmids, and oligonucleotides used and constructed in this study are described in Tables S4 to S8. *P. aeruginosa* strains were grown at 37°C or 28°C, in L broth (LB; 1% bactotrypton, 0.5% yeast extract, 0.5% NaCl), supplemented with antibiotics when

necessary: 75 $\mu$g mL$^{-1}$ chloramphenicol, carbenicillin at 300 $\mu$g mL$^{-1}$, rifampicin at 300 $\mu$g mL$^{-1}$, or M9 medium (94) supplemented with 100 $\mu$g mL$^{-1}$ leucine and 0.25% citrate or 0.5% glucose. IPTG (isopropyl-$\beta$-D-thiogalactopyranoside) at indicated concentrations was used for *tac*p induction. Allele exchange in the *P. aeruginosa* chromosome was conducted with the use of suicide vector pAKE600 (95) as described in the supplemental material.

**Protein–protein interactions analyses.** The bacterial two-hybrid system, based on the reconstitution of *Bordetella pertussis* adenylate cyclase (CyaA), was used (51). *E. coli* BTH101 *cya-* were transformed with pairs of complementary vectors containing *cyaT18* and *cyaT25* fragments fused with indicated genes (fragments). Transformants were selected on McConkey agar base (BD, 281810) supplemented with 1% maltose, appropriate antibiotics, and 0.15 mM IPTG. After 3 days at 28°C, random colonies were restreaked to a fresh plate, incubated for 2 days, photographed, and used to inoculate cultures (LB with antibiotics, 0.15 mM IPTG) grown overnight at 28°C before performing $\beta$-galactosidase activity assays (96). A library containing random genomic DNA fragments of PAO1161 fused to the C-terminal part of CyaT18 was prepared by fragmentation of PAO1161 genomic DNA by nebulization, blunting of DNA with a mixture of T4 DNA polymerase and Klenow fragment, and ligation with SmaI digested pUT18C (51). The library consists of plasmids from around 250,000 colonies, with 40% of colonies containing an insert and an average insert size of 800 bp, which overall yields a 10× coverage of the genome. Identification of ParB partners using BACTH was performed by transformation of the library into *E. coli* BTH101 cells expressing CyaT25-ParB. Cells were plated on MacConkey medium and incubated at 28°C for 4 days, and red colonies were restreaked. Following plasmid isolation and retransformation, inserts of colonies showing red appearance on MacConkey plates were sequenced (Table S2).

To identify ParB partners with coimmunoprecipitation, *Pae* strain with *flag-parB* was used along with WT control strain. Exponentially growing cells (OD$_{600}$ ~0.8) were suspended in lysis buffer (10 mM Tris-HCl, pH = 8.0, 20% sucrose, 40 mM EDTA, 0.5 mg mL$^{-1}$ lysozyme) and incubated 30 min on ice. Subsequently, 1 volume of 2× IP buffer (1.5 M Tris, pH = 7.0, 0.3 M NaCl, 0.2% Triton X-100) was added along with PMSF (phenylmethylsulfonyl fluoride; final concentration 1 mM) and protease inhibitor cocktail (P8465, Sigma, diluted 100×), and cells were disrupted by sonication. Clarified lysates were incubated with Dynabeads (14311D, Thermo) coupled with anti-FLAG antibodies (MA1-91878, Thermo) and washed with 1× IP buffer, and proteins were eluted from the beads by sequential vortexing for 5 min at 22°C after addition of 0.1 M Glycine (pH 3.5). pH of the samples was neutralized, and proteins in fractions with highest content of FLAG-ParB (as judged by Western blotting), and corresponding control samples, were identified by mass spectrometry analysis (Mass Spectrometry Laboratory IBB PAS). Data were analyzed using MascotScan software (Matrix Science, Inc., USA). Analyses of ParB interactions with partners upon their expression in a heterologous host with the use of coimmunoprecipitation were conducted as described previously for the analysis of ParB–ParA interactions (36). *E. coli* BL21 cells transformed with pABB1.2 and pET28a derivatives encoding ParB and partners, respectively, were used. Samples were subjected to Western Blot analysis with use of anti-His$_6$ antibodies.

**Fluorescence microscopy and image analysis.** Cells expressing fluorescent protein fusions were cultured and fixed by addition of 1 volume of 2.8% formaldehyde, 0.04% glutaraldehyde. For DNA staining, cells were harvested and fixed in formaldehyde, followed by fixation in EtOH, overnight incubation in solution containing RNase, and staining with propidium iodide/SYTO9 mixture. Fluorescence microscopy images were captured using a Zeiss Imager M2 and analyzed using ImageJ and MicrobeJ (97). Detailed microscopy protocols and description of image quantification are included in the supplemental material.

**Chromatin immunoprecipitation and sequencing (ChIP-seq).** ChIP-seq analysis was performed essentially as described before (39). Polyclonal antibodies against ParB and cells growing exponentially in LB medium (OD$_{600}$ 0.4 to 0.5) were used. Reads quality controlled using fastp (98) were mapped to the *P. aeruginosa* PAO1161 genome (accession number CP032126 [80]) using Bowtie (v.2.3.4.3 [99]). The presented data represent the ratio between the number of reads mapping to the *parS1-4* ± 15 kbp region and reads mapping to regions encompassing half-*parS* sequences, defined as range of peaks called for combined WT replicates using MACS2 v.2.1.2 (100) with fold enrichment of >3 (Table S3 in the supplemental material). Remaining parts of the genome were used to calculate the theoretical coverage background, which was subtracted from both values, before calculating the ratio. The read coverage of half-*parS* containing regions provides an internal control for each sample, correcting for unequal ChIP efficiency. Calculation of the number of reads mapping to indicated genome parts in ChIP samples was done using the plotEnrichement function from deeptools (v.3.3.0 [101]).

**Data availability.** Raw sequencing data were deposited in the NCBI's Gene Expression Omnibus (GEO) database (http://www.ncbi.nlm.nih.gov/geo/) under accession number GSE213881.

## SUPPLEMENTAL MATERIAL

Supplemental material is available online only.
**SUPPLEMENTAL FILE 1**, XLSX file, 0.06 MB.
**SUPPLEMENTAL FILE 2**, XLSX file, 0.01 MB.
**SUPPLEMENTAL FILE 3**, XLSX file, 0.02 MB.
**SUPPLEMENTAL FILE 4**, PDF file, 2.6 MB.

## ACKNOWLEDGMENTS

We thank Marta Elżbieta Kotlarek, Anna Agnieszka Stańczyk, and Olga Sokolnicka for construction of several plasmids used in this study; and Jakub Piątkowski (Institute of Genetics and Biotechnology, University of Warsaw) and Małgorzata Łobocka (IBB PAS) for providing plasmids used in this work. This work was funded by the National Science Centre in Poland, grants 2013/11/B/NZ2/02555 and 2018/29/B/NZ2/01745 obtained by G.J.-B.

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
