## [Reviewer comments · Microbiology Spectrum]

Microbiology Spectrum

Diverse partners of the partitioning ParB protein in *Pseudomonas aeruginosa*

Adam Kawalek, Krzysztof Glabski, Aneta Bartosik, Dominika Wozniak, Magdalena Kusiak, Jan Gawor, Karolina Zuchniewicz, and Grazyna Jagura-Burdzy

Corresponding Author(s): Grazyna Jagura-Burdzy, Instytut Biochemii i Biofizyki Polskiej Akademii Nauk

Review Timeline:

Submission Date:	October 21, 2022
Editorial Decision:	November 21, 2022
Revision Received:	December 2, 2022
Accepted:	December 16, 2022

Editor: Elitza Tocheva

Reviewer(s): The reviewers have opted to remain anonymous.

Transaction Report:

DOI: <https://doi.org/10.1128/spectrum.04289-22>

November 21, 2022

Prof. Grazyna Jagura-Burdzy
Institute of Biochemistry and Biophysics, Polish Academy of Sciences
Pawinskiego 5A
Warsaw 02-106
Poland

Re: Spectrum04289-22 (**Diverse partners of the partitioning ParB protein in *Pseudomonas aeruginosa***)

Dear Prof. Grazyna Jagura-Burdzy:

Link Not Available

Sincerely,

Elitza Tocheva

Journals Department
Reviewer comments:

Reviewer #1 (Comments for the Author):

In this manuscript, Kawalek et al identified the N-terminal peptide on *P. aeruginosa* ParB that mediates interaction with its partner ParA. This is not so surprising because the N-terminus of ParB has been known to harbor this role in multiple bacterial species, but Kawalek et al have alanine-scanned to characterize in further depth this region, so there is new findings here. Kawalek et al then combined co-IP and BACTH to identify putative binding partners of *P. aeruginosa* ParB, they have identified multiple partners however the biological significance of the interaction between ParB and these putative binding partners is entirely unclear.

Kawalek et al repeatedly wrote "ParB interactions with ParA ...prevents DNA entangling by SMC". The use of the word

"entangle" here is wrong/misleading. Entanglement often means DNA intertwines together randomly. SMC is known to prevent DNA entanglement, not promote it. I assume Kawalek et al mean "cohesion" instead of "entanglement", that SMC potentially coheses the 2 replicated sister chromosomes more when oriC-segregation by ParB-ParA is impaired. I would suggest using the word "coheses/cohesion" throughout the manuscript.

Minor comments:

Line 72: PopZ is not an ATPase?

Line 99 Replace "CTP processing" with CTP binding and hydrolysis?

Line 302: add the word "putative" as in "we have identified putative proteins interacting with partitioning protein parB..."

Reviewer #2 (Comments for the Author):

ParAB-parS plays an important role in the faithful segregation of chromosomes. Kawalek et al have shown that impaired interaction of ParB and Par A proteins leads to growth retardation and increased frequency of anucleated cells. Further, smc deleted strains have been shown to reduce the detrimental effect of disruption of ParA-ParB interaction. Proteomics techniques have been used to identify the novel ParB interacting proteins that includes enzymes, nucleoid-associated proteins, and stress proteins etc. As such, the present study provides new insight into the role of ParA-ParB interaction in chromosome segregation and also identifies novel interacting partners of ParB. However, a triple deletion strain of parB, smc and mksB would have provided a deeper understanding of the genetic interaction of parABS with condensins. Overall, the manuscript is well written, and the experiments performed to support the conclusions. However, the authors should address the following points to strengthen their findings.

1. Bartosik et al 2004, have shown that ParB C-terminal domain interacts with ParA by yeast two hybrid assay; on the contrary, the present study shows ParB N-terminal interacts with ParA using BACTH system. Therefore, it may be required to validate the aforementioned interactions in-vitro or discuss this discrepancy in detail.

2. THE BACTH assay results suggest that point mutants of ParB do not interact with ParA. However, it is possible that point mutants still may interact with ParA, albeit with low affinity when compared to wild-type ParB. Is the BACTH assay sensitive enough to detect such weak interactions?

3. Do the point mutations in the N-terminal domain of ParB also affect its oligomeric state in solution and its DNA binding property?

4. smc deletion is known to cause defects in chromosome segregation, especially in rich media. Therefore, the effect of ParB in chromosome partitioning in smc null strain of *P. aeruginosa* should also be studied in such conditions.

5. Authors can discuss how ParB interacting proteins would influence the ParA-ParB interaction or DNA binding activity of ParB.

Staff Comments:

Preparing Revision Guidelines

Please return the manuscript within 60 days; if you cannot complete the modification within this time period, please contact me. If you do not wish to modify the manuscript and prefer to submit it to another journal, please notify me of your decision immediately so that the manuscript may be formally withdrawn from consideration by Microbiology Spectrum.

Comments for the authors:

ParAB-*parS* plays an important role in the faithful segregation of chromosomes. Kawalek et al have shown that impaired interaction of ParB and Par A proteins leads to growth retardation and increased frequency of anucleated cells. Further, *smc* deleted strains have been shown to reduce the detrimental effect of disruption of ParA-ParB interaction. Proteomics techniques have been used to identify the novel ParB interacting proteins that includes enzymes, nucleoid associated proteins, and stress proteins etc. As such, the present study provides new insight into the role of ParA-ParB interaction in chromosome segregation and also identifies novel interacting partners of ParB. However, a triple deletion strain of *parB*, *smc* and *mksB* would have provided deeper understanding on the genetic interaction of *parABS* with condensins. Overall, the manuscript is well written and the experiments performed support the conclusions. However, authors should address the following points to strengthen their findings.

1. Bartosik et al 2004, have shown that ParB C-terminal domain interacts with ParA by yeast two hybrid assay, on the contrary the present study shows ParB N-terminal interacts with ParA using BACTH system. Therefore, it may be required to validate the aforementioned interactions *in-vitro* or discuss this discrepancy in detail.
2. THE BACTH assay results suggest that point mutants of ParB do not interact with ParA. However, it is possible that point mutants still may interact with ParA, albeit with low affinity when compared to wild-type ParB. Is the BACTH assay sensitive enough to detect such weak interactions?
3. Do the point mutations in the N- terminal domain of ParB also affect its oligomeric state in solution and its DNA binding property?
4. *smc* deletion is known to cause defects in chromosome segregation, especially in rich media. Therefore, the effect of ParB in chromosome partitioning in *smc* null strain of *P aeruginosa* should also be studied in such conditions.
5. Authors can discuss how ParB interacting proteins would influence the ParA-ParB interaction or DNA binding activity of ParB.

Reviewer comments:

Reviewer #1 (Comments for the Author):

In this manuscript, Kawalek et al identified the N-terminal peptide on P. aeruginosa ParB that mediates interaction with its partner ParA. This is not so surprising because the N-terminus of ParB has been known to harbor this role in multiple bacterial species, but Kawalek et al have alanine-scanned to characterize in further depth this region, so there is new findings here. Kawalek et al then combined co-IP and BACTH to identify putative binding partners of P. aeruginosa ParB, they have identified multiple partners however the biological significance of the interaction between ParB and these putative binding partners is entirely unclear.

This work aimed for identification of the putative partners of Par proteins. We agree that the role of the most of them, (with the exception of the Min system although still not thoroughly investigated in *P. aeruginosa*), is unclear and awaits further studies focused on the particular partners. The significance of this study is in the demonstration of the wide spectrum of ParB partners, and hence processes within the cell, which can be linked with the process of DNA segregation.

Kawalek et al repeatedly wrote "ParB interactions with ParA ...prevents DNA entangling by SMC". The use of the word "entangle" here is wrong/misleading. Entanglement often means DNA intertwines together randomly. SMC is known to prevent DNA entanglement, not promote it. I assume Kawalek et al mean "cohesion" instead of "entanglement", that SMC potentially coheses the 2 replicated sister chromosomes more when oriC-segregation by ParB-ParA is impaired. I would suggest using the word "coheses/cohesion" throughout the manuscript.

We changed the 'entanglement' word to 'cohesion' throughout the manuscript.

Minor comments:

Line 72: PopZ is not an ATPase?

We apologize for this mistake. Corrected now (line 75).

Line 99 Replace "CTP processing" with CTP binding and hydrolysis?

Corrected as suggested (line 102).

Line 302: add the word "putative" as in "we have identified putative proteins interacting with partitioning protein parB..."

Corrected as suggested (line 307)

Reviewer #2 (Comments for the Author):

ParAB-parS plays an important role in the faithful segregation of chromosomes. Kawalek et al have shown that impaired interaction of ParB and ParA proteins leads to growth retardation and increased frequency of anucleated cells. Further, smc deleted strains have been shown to reduce the detrimental effect of disruption of ParA-ParB interaction. Proteomics techniques have been used to identify the novel ParB interacting proteins that includes enzymes, nucleoid-associated proteins, and stress proteins etc. As such, the present study provides new insight into the role of ParA-ParB interaction in chromosome segregation and also identifies novel interacting partners of ParB. However, a triple deletion strain of parB, smc and mksB would have provided a deeper understanding of the genetic interaction of parABS with condensins.

The genetic interaction between *par* and *mks* genes in *P. aeruginosa* has been extensively investigated previously by Lioy and colleagues (ref [48]), as well as Zhao and colleagues (ref [49]). According to the first study, a $\Delta parS$ (strain with mutated *parS1-4* sites) $\Delta mksB$ can be constructed, however the amount of anucleate cells in the cultures of such mutant reaches 24% whereas $\Delta parS$ strain produces 3% and Δmks or WT <0.15%. The $\Delta mks \Delta smc$ cultures contained 0.5% of anucleate cells. In the second study the authors reported around 3% of anucleate cells for a $\Delta parB$ strain, around 1% for $\Delta mksB$, Δsmc , or $\Delta mksB \Delta smc$ strains, whereas more than 16% was observed in case of $\Delta mksB \Delta parB$ strain.

We attempted the construction of a double deletion strain $\Delta mksBEF \Delta parAB$ by replacing *parAB* in an $\Delta mksBEF$ background with an antibiotic resistance cassette, however we repeatedly failed to obtain a viable mutant, so we could not proceed with the construction of the triple mutant. Perhaps less stringent conditions (temperature, concentration of antibiotic) are required for the selection of such strongly impaired mutant strain. Nevertheless we think that the above-mentioned studies explored this subject sufficiently, and showed that the MksBEF contribution to the segregation process becomes critical in the absence of the ParAB-*parS* system. It is worth to point out that in the current study we have used $\Delta mksBEF$ background to eliminate the ‘backup’ DNA segregation pathway, which we were hoping would allow us to see phenotypic defects of the constructed strains, if absence of the selected ATPases would have impaired functioning of the ParAB-*parS* system.

Overall, the manuscript is well written, and the experiments performed to support the conclusions. However, the authors should address the following points to strengthen their findings.

1. Bartosik et al 2004, have shown that ParB C-terminal domain interacts with ParA by yeast two hybrid assay; on the contrary, the present study shows ParB N-terminal interacts with ParA using BACTH system. Therefore, it may be required to validate the aforementioned interactions in-vitro or discuss this discrepancy in detail.

The observation that ParB C-terminal domain interacts with ParA by yeast two hybrid (YTH) assay was puzzling at that time. Whereas self-association of ParB could be observed as strong interactions in YTH, the interactions between ParB and ParA in YTH were much weaker. At that time, the understanding of possible YTH system artifacts was not so advanced, and in the next years multiple sources of artifacts of the YTH data were summarized (e.g. by Manfred Koegl, Peter Uetz 2008, <https://doi.org/10.1093/bfpg/elm035>). The vectors that were used in our study were based on pBTM116 encoding bacterial LexA repressor (DNA binding domain) and pGAD424 encoding activation domain of yeast Gal4 protein. They allowed only fusion of the investigated proteins from their N termini. Our data with BACTH demonstrated clearly that ParA needs to have N-termini free for efficient interactions with ParB (ref [36]), suggesting that the reported YTH result could be a false positive. It’s possible that observed interactions between C-terminus of ParB and ParA could be caused by very strong dimerization ability of this part of ParB, leading to self-assembly of LexA-ParB fusion proteins in the form of nucleoprotein complex, that changed topology of DNA around promoters with multiple LexA operators and facilitated transcription activation by Gal4. Since we can only speculate about the cause of different results in the yeast- and bacterial- two hybrid systems we do not discuss this in detail in the manuscript (and also because of space constraints). Suggestion that the YTH result should be considered as an artifact has been added to the revised version of the manuscript (lines 112-114)

2. THE BACTH assay results suggest that point mutants of ParB do not interact with ParA. However, it is possible that point mutants still may interact with ParA, albeit with low affinity when compared to wild-type ParB. Is the BACTH assay sensitive enough to detect such weak interactions?

We do consider such a possibility, as suggested previously in the discussion (lines 334-335). We have modified the results section to highlight this limitation of BACTH assay more clearly (lines 119-122). Nevertheless the final conclusion is that the single amino acids substitutions in the N-terminus of ParB have an impact on the ability to interact with ParA.

3. Do the point mutations in the N-terminal domain of ParB also affect its oligomeric state in solution and its DNA binding property?

We think that there is very limited (if any) influence of the analyzed ParB substitutions in the N-terminal part on the protein oligomerization and DNA (*parS*, half-*parS*) binding properties. This is based on the following observations (reported in the manuscript or previously published by us):

1) ParB G11A binding and spreading around *parS*s (and binding to half-*parS*s) was observed in ChIP-seq (ChIP-seq data available under accession number GSE213881) and no major effect of this amino acid substitution on the extent of ParB spreading (Fig. 2G).

2) N-terminally truncated ParB (Δ 1-18) showed the same ability to silence the gene expression in the test plasmid as the full length ParB protein (Table 3, ref [45]). In the silencing test, overproduced ParB from one plasmid binds to *parS* on the compatible plasmid, spreads on DNA and forms a nucleoprotein complex leading to the reduced expression of genes required for plasmid replication. This is observed as a 1000-fold reduction in the efficiency of transformation with two plasmids. ParB variants with impaired DNA binding, dimerization or carrying substitutions in the N-terminal ParB domain (responsible for CTP binding and hydrolysis) do not demonstrate the gene silencing effect in this test (ref. [40], [54]). ParB variants with alanine substitutions at position 8 to 12 were analyzed with this test, and they did not differ from the WT ParB protein (data not shown).

3) Analyzed ParB variants formed foci when fused to YFP (Fig. 2 and Figure S2). ParB foci formation by *P. aeruginosa* ParB was shown to rely on *parS* binding and spreading on the DNA (ref [35], [40]).

These observations convincingly suggest that the L8A, R10A, G11A, L12A substitutions do not have impact on ParB spreading on DNA, which relies on ParB dimerization, oligomerization and DNA binding. Since we have not seen an effect on the ParB functioning *in vivo* we have not analyzed these properties of ParB variants *in vitro*.

4. smc deletion is known to cause defects in chromosome segregation, especially in rich media. Therefore, the effect of ParB in chromosome partitioning in smc null strain of P aeruginosa should also be studied in such conditions.

Lioy and colleagues (ref [48]) reported no differences in anucleate cells content in the Δ *smc* strain relative to WT (<0.15%) when grown in mineral medium with citrate or glucose/casamino acids at 30°C, or LB at 30°C. Content of anucleate cells was slightly higher when Δ *smc* cells were grown in LB at 37°C (Δ *smc* < 1%, WT <0.15%, Supplementary Figure 1 in ref [48]). Zhao and colleagues (ref [49]) observed around 1% anucleate cells in Δ *smc* cultures (WT <0.2%) for cells grown in M9 with citrate at 37°C. There are other reports showing that Δ *smc* mutants of PAO1 produced 2.5% anucleate cells both in LB and in M9 with glucose grown at 37°C (Petrushenko et al., 2011, doi: 10.1111/j.1365-2958.2011.07763.x.) or ~1% (Zhao et al., 2016, <https://doi.org/10.1128/JB.00448-16>). This indicates that there is no major influence of SMC on the segregation process in *P. aeruginosa* as judged by the analysis of anucleate cells content alone.

In our hands in PAO1161 (a PAO1 derivative – discussion of the differences between the strains in ref [80]), the Δ *smc* strain cultures do not contain significantly more anucleate cells than WT when grown in rich LB medium at 37°C, (doubling time 30±1 min) (Fig. 2E). The same is observed in M9 medium containing glucose (37°C) or citrate (28°C), conditions in which WT *P. aeruginosa* cultures have reduced doubling time (e.g. 110 min in case of medium with glucose) (Fig. S4AB). Of course, we have only looked at the amount of anucleate cells and there are other defects, which were

recently described in an elegant study by Liroy et al., (ref [48]), thus we did not analyze the phenotype of Δsmc mutant in detail as a part of this manuscript.

5. Authors can discuss how ParB interacting proteins would influence the ParA-ParB interaction or DNA binding activity of ParB.

The influence of partners on e.g. the extent of ParB spreading is indeed something to be investigated. Concomitantly, our data (Fig 5A) indicate that, at least the four putative NTPases, are unlikely to compete with ParA for ParB binding, as they interact with truncated ParB 37-290 variant lacking the motif shown to be the ParA binding region. The molecular basis of the interaction between ParB and selected partners is currently investigated and will be included in another manuscript. The discussion has been modified to suggest the possible impact of partners on the indicated ParB properties (line 395).

December 10, 2022

Prof. Grazyna Jagura-Burdzy
Instytut Biochemii i Biofizyki Polskiej Akademii Nauk
Pawinskiego 5A
Warsaw 02-106
Poland

Re: Spectrum04289-22R1 (**Diverse partners of the partitioning ParB protein in *Pseudomonas aeruginosa***)

Dear Prof. Grazyna Jagura-Burdzy:

Your manuscript has been accepted, and I am forwarding it to the ASM Journals Department for publication. You will be notified when your proofs are ready to be viewed.

Sincerely,

Elitza Tocheva
Editor, Microbiology Spectrum

Journals Department
Table S2: Accept
Supplemental Material: Accept
Table S1: Accept
Table S3: Accept